# Evaluation Methods Applied to Digital Health Interventions: What Is Being Used beyond Randomised Controlled Trials?—A Scoping Review

**DOI:** 10.3390/ijerph19095221

**Published:** 2022-04-25

**Authors:** Robert Hrynyschyn, Christina Prediger, Christiane Stock, Stefanie Maria Helmer

**Affiliations:** 1Institute of Health and Nursing Science, Charité–Universitätsmedizin Berlin, Corporate Member of Freie Universität Berlin and Humboldt-Universität zu Berlin, Augustenburger Platz 1, 13353 Berlin, Germany; christina.prediger@charite.de (C.P.); christiane.stock@charite.de (C.S.); sthelmer@uni-bremen.de (S.M.H.); 2Unit for Health Promotion Research, Department of Public Health, University of Southern Denmark, Degnevej 14, 6705 Esbjerg, Denmark; 3Human and Health Sciences, University of Bremen, Grazer Strasse 4, 28359 Bremen, Germany; 4Leibniz Science Campus Digital Public Health, 28359 Bremen, Germany

**Keywords:** digital health, evaluation methods, complex interventions, scoping review

## Abstract

Despite the potential of digital health interventions (DHIs), evaluations of their effectiveness face challenges. DHIs are complex interventions and currently established evaluation methods, e.g., the randomised controlled trial (RCT), are limited in their application. This study aimed at identifying alternatives to RCTs as potentially more appropriate evaluation approaches. A scoping review was conducted to provide an overview of existing evaluation methods of DHIs beyond the RCT. Cochrane Central Register of Controlled Trials, MEDLINE, Web of Science, and EMBASE were screened in May 2021 to identify relevant publications, while using defined inclusion and exclusion criteria. Eight studies were extracted for a synthesis comprising four alternative evaluation designs. Factorial designs were mostly used to evaluate DHIs followed by stepped-wedge designs, sequential multiple assignment randomised trials (SMARTs), and micro randomised trials (MRTs). Some of these methods allow for the adaptation of interventions (e.g., SMART or MRT) and the evaluation of specific components of interventions (e.g., factorial designs). Thus, they are appropriate for addressing some specific needs in the evaluation of DHIs. However, it remains unsolved how to establish these alternative evaluation designs in research practice and how to deal with the limitations of the designs.

## 1. Introduction

Digitalisation is making inroads into healthcare and has been amplified by the COVID-19 pandemic, associated with digital contact tracking measures and digital video consultation opportunities [1]. Digital health interventions (DHIs) have the potential to improve the health of individuals by delivering health promotion interventions and support for behavioural change [2]. As scalable digital tools, they can be equitably shared across populations with different health needs [3]. DHIs are characterised by their ability to support and serve different health needs of providers, patients, and populations formally or informally through digital technologies. The application and use of DHIs is diverse and ranges from simple SMS support to complex modular interventions that can be used as an app for doctors, patients, or entire populations [4]. Due to the high utilization of technology worldwide, DHIs are becoming particularly relevant for the delivery of health promotion, prevention, and health care services. This is underscored by the growth of mobile devices. While approximately 2.5 billion people worldwide used a smartphone in 2016, the number of users is expected to increase to approximately 4.5 billion by 2024 [5]. Despite the increase in use and expectations, scientific evaluation practice is often lagging behind and cannot keep up with the pace of technology [6]. 

Similar to pharmaceuticals, it is important for DHIs to use reliable evaluation methods to assess the efficacy of these interventions. Randomised controlled trials (RCTs) represent the gold standard for the evaluation of interventions. They count as the main research design for demonstrating causal relationships between intervention and outcome [7]. Randomisation and rigidly controlling the environment allow RCTs to minimize selection bias and confounding factors [8].

However, with the onset of digitisation in medicine, health promotion, and prevention challenges regarding the appropriateness of RCTs as evaluation design are also becoming apparent. First, the long timeframe of RCTs is seen as a core problem in the evaluation of DHIs, as technology can be outdated when the evaluation of its efficiency has been completed [9]. Second, the rigidity of RCT study protocols is described as incompatible for DHIs, as these protocols often have a flexible and context-dependent focus [10]. Due to this flexible and context-dependent focus, DHIs are regarded as “complex” interventions [11]. Third, DHIs have various specificities (e.g., interaction effects, attrition rates) that RCTs can insufficiently address. For example, DHIs are used by individuals but they are evaluated in between-group study designs. DHIs need to meet individual needs, so it is plausible to assume that they need to be tailored individually to maximise their effectiveness [12]. Individual tailoring also needs to incorporate user preferences, so that acceptability and engagement is ensured [13]. 

While the RCT may not be inappropriate for the effect evaluation of all DHIs, alternative evaluation approaches should also be considered to assess and evaluate DHIs as they might be more suitable to address specific needs of DHIs.

The British Medical Research Council proposed extending the range of study designs for evaluation beyond RCTs and mentioned, e.g., stepped-wedge designs or N-of-1 designs that can be considered as alternative options [14]. These alternative options for the evaluation of interventions were summarized under the heading of “alternative study designs”. The definition alternative study designs was also adopted from other authors [15].

Alternative study designs or adaptive study designs describe a procedure in which different decisions are made regarding measurement, dosage, and time that affect the intervention [16,17]. In the literature, these approaches are often referred to as dynamic treatment regimes [18]. Prominent examples of adaptive study designs are the sequential multiple assignment randomised trial (SMART) and multiphase optimisation strategy (MOST), including factorial designs to measure effects [19]. 

Although adaptive study designs and their suitability for the alternative evaluation of DHIs are discussed intensely in the scientific literature, it seems that there are only a few studies conducted that actively performed these adaptive study designs [20]. Within the present scoping review, a horizon scanning of scientific literature was conducted with the primary aim to identify alternative evaluation methods that are implemented to evaluate DHIs in healthcare as well as in health promotion and prevention. A second aim was to provide insights into their respective advantages and limitations. Thereby, this review contributes to a better understanding of an emerging field of research and can guide future evaluation practice of digital interventions within health promotion, prevention, and healthcare.

## 2. Materials and Methods

A scoping review has been conducted to present a broad overview of evidence, to identify key concepts that are used in published research, and which knowledge gaps are existing [21]. Results were reported according to PRISMA guidelines for scoping reviews [22]. The filled-in Preferred Reporting Items for Systematic reviews and Meta-Analyses extension for Scoping Reviews (PRISMA-ScR) can be found in the Appendix A. The Arksey and O’Malley [23] methodology framework for scoping reviews guided the conduct of this review and following this framework no quality assessment of the studies was intended.

### 2.1. Inclusion and Exclusion Criteria

The inclusion and exclusion criteria were determined a priori in a consensus process by the researchers conducting the review (S.M.H and R.H.). The following inclusion criteria have been used to identify relevant literature regarding the research question. Studies were included, that were (1) applying alternative evaluation methods, (2) testing and reporting effects of interventions, and (3) dealing with DHIs. Inclusion was not restricted to any specific population or to specific contexts in which studies were conducted.

Exclusion criteria were (1) randomised controlled trials, pragmatic randomised controlled trials, formative evaluation studies, and pre-post studies without control condition, (2) studies that merely provided telephone support, and (3) studies that were published in languages other than German or English. 

Conference proceedings, abstracts without full texts, and study protocols without reporting results were excluded from this review. Moreover, systematic reviews were excluded but it was checked whether they included original studies that can be of interest for the review.

### 2.2. Literature Search

The search was conducted in May 2021 in the databases Cochrane Central Register of Controlled Trials (CENTRAL), MEDLINE, Web of Science, and EMBASE by one author (R.H.). The search terms “digital health” and “evaluation methods” were used and enriched with synonyms, truncations, and Medical Subject Headings (MeSH). The search terms were informed by an earlier review of Pham et al. [20]. Table 1 provides an overview of the search strategy. The following search syntax was used in the databases: 

((“telemedicine” [MeSH Terms] OR “telemedicine” [All Fields] OR “tele-medicine” [All Fields] OR “telehealth” [All Fields] OR “tele-health” [All Fields] OR “mhealth” [All Fields] OR “m-health” [All Fields] OR “ehealth” [All Fields] OR “e-health” [All Fields])) AND (“Research Design” [MAJR] OR “Evaluation Studies as Topic” [Mesh:NoExp] OR research method*[tiab] OR research strateg*[tiab] OR methodolog*[tiab]) AND (alternative*[tiab] OR effective*[tiab] OR evaluation*[tiab] OR “Multiphase optimisation strategy” OR “Factorial design” OR “micro-randomisation” OR “Adaptive treatment strategies” OR “Sequential Multiple Assignment Randomised Trial” OR “n-of-1 trials” OR “n 1 trials” OR (“n-of-1” AND “trial”))

The author (R.H.) conducted a comprehensive search that was limited neither by publication date nor publication type or any other filters. Reference lists of the included full texts were scanned for potentially relevant other articles.

The selection of sources of evidence was executed in two iterative steps after all identified records had been exported to EndNote X9.1 (Clarivate Analytics, Philadelphia, PA, USA) and Rayyan (Rayyan Systems Inc., Cambridge, MA, USA). First, three authors (R.H., S.M.H, C.P.) selected eligible studies by screening titles and abstracts of all identified records independently, using the web application Rayyan. If the title and abstract of the respective study did not seem appropriate for the research question, it was excluded. If there was uncertainty or a final assessment of the study was not possible, the study was included to avoid excluding important studies in the first screening phase. Second, full texts of all records that were deemed relevant after the first screening phase were obtained and reviewed by two authors (R.H., S.M.H) for eligibility. Any discrepancies were resolved via discussion, or, in the case of no consensus, a third reviewer (C.P.) was involved. Data from all included studies were independently extracted by two reviewers (R.H., S.M.H.) into a Microsoft Excel 2019 (Microsoft Corporation, Redmond, WA, USA) table developed for the research purpose to capture all relevant information. The extraction table consisted of the following segments: country of intervention, target condition of the DHI, number of included cases, investigation period, and source of funding. Further, for the report, interventions were sorted according to the characteristics of the specific study designs and their difference to RCTs.

## 3. Results

In total, 5603 publications were identified, of which eight studies were included by hand research or citation searching. After removing duplicates and applying inclusion criteria to the full texts of eligible articles (*n* = 176), eight studies were included in the scoping review that met all pre-defined inclusion criteria but were not disqualified based on exclusion criteria during the iterative screening phase. A PRISMA flow diagram for study selection is presented in Figure 1.

### 3.1. Description of the Study Characteristics 

Studies with alternative study designs were limited to three world regions (see Table 2). Most studies (75.0%) were conducted in North America (USA) [25,26,27,28,29,30]. One study (12.5%) was conducted in Europe (Belgium) [31] and one (12.5%) in South America (Brazil) [32]. 

Various target conditions were represented, with the area of physical activity being mostly addressed (62.5%) [25,26,27,31,32]. One study each focussed on smoking cessation [30], weight loss [29], and chronic pain in children [28].

Five (62.5%) of the included studies reported funding from public sources [25,26,30,31,32]. In one (12.5%) study, funding was unclear [29] and two (25.0%) studies reported mixed funding consisting of public and private funding [27,28].

The study duration (including follow-ups) ranked from five weeks [31] to 24 weeks (mean: 16 weeks; median: 18 weeks) [29,30,32]. No study reported a duration shorter than one month, but three studies [25,27,31] were conducted in less than three months.

The sample size varied widely between included studies. The smallest sample size was involved in a Sequential multiple assignment randomisation trial [32] with 18 participants, whereas 1866 participants were included in a fractional factorial randomised trial [30]. The mean sample size was 416 and the median 134. 

In total, the eight included studies identified four alternative study designs, which will be further discussed according to their difference to RCTs: Micro randomisation trial [25],(Fractional) Factorial randomised controlled trial [26,27,29,30,31],Sequential multiple assignment randomisation trial [32],Stepped-wedge cluster randomised trial [28].

Only two studies (25.0%) reported an exclusive control condition. In a sequential multiple assignment randomised trial, a passive control group was used [32] and a stepped-wedge cluster randomised trial reported usual care at the clinic level as the control condition [28]. The other alternative study designs reported no exclusive control condition as a comparison of different intervention components. 

### 3.2. Difference of Identified Studies Compared to RCTs

#### 3.2.1. Micro Randomisation Trial

The aim of the study by Klasnja et al. [25] was to evaluate the efficacy of a mHealth intervention named “HeartSteps” in order to optimise the intervention. The authors conducted a MRT of a just-in time adaptive intervention (JITAI). MRTs are particularly used to optimise JITAI. Therefore, the intervention components are randomised for each participant each time the system (in this case, the mHealth app) provides the components. This design enables the modelling of causal effects of each intervention component and the simultaneous modelling of intervention effects over time by assessing the outcomes repeatedly [25].

In contrast to RCTs, MRTs are intended to evaluate the individual intervention components and therefore intervention delivery starts immediately after the initial patient inclusion. A baseline assessment is not existing. Another aspect that is relevant for MRTs is the randomisation process, which took place for each participant at five decision points in the study. The randomisation in MRTs is independently implemented of prior randomisations and participants’ responses. Although the study duration, compared to the other study designs, was short (42 days), it was possible that each participant was randomised up to 210 times and therefore received different individual intervention components. The high number of randomisations is also reflected in the data collection time points. While the data collection is limited to baseline and post-test in an RCT, the study design shows continuous data collection over the duration of the study. This resulted in up to 7540 data collection time points in the study. Furthermore, while RCTs usually include at least one control group to attribute effects to the intervention, the MRT does not. Here, the focus is on comparisons of the different intervention components. These are tested for effectiveness regarding the outcomes to subsequently exclude ineffective components [25].

#### 3.2.2. (Fractional) Factorial Randomised Controlled Trials

Most of the included studies were conducted as factorial randomised trials [26,27,29,31]. All factorial randomised trials were designed to test the efficacy of a multicomponent intervention. For example, Adams et al. [26] evaluated the effects of goal setting and rewards to increase steps walked per day. This study was based upon a previous study [33] in which it remained unclear whether the previously observed effects were attributable to goal setting or reward components. Factorial trials can provide an efficient way for untangling multicomponent interventions and theoretical mechanisms [34]. The authors of the studies stated that factorial designs help to identify active components of DHIs for driving behavioural change or developing optimised treatment packages [26,27,29,31]. 

The designs of factorial trials can be numerous. For example, this scoping review identified two 2 × 2 factorial designs, one 2 × 5 factorial design, one 2 × 2 × 2 factorial design, and one 2 × 4 fractional factorial design. The different factors and levels which make the design complex are dependent on the different functions and utilities of the DHI. This complexity is well demonstrated in the study by Strecher et al. [30], as this was the only identified study that conducted a fractional factorial trial. While a full factorial trial of the study by Strecher et al. [30] would have had 32 intervention arms, the implementation of the fractional factorial design allows reducing the intervention arms to 16. These multicomponent analyses and flexible designs were most commonly used for DHIs in this review [35]. 

In contrast to RCTs which aim to measure effects of the overall intervention as a result, (fractional) factorial designs allow for analysing which mix of intervention component is the most effective. However, because DHIs are often developed as multicomponent or modular structures, RCTs seem to fall short here and (fractional) factorial designs can be seen as an efficient possibility to compare different intervention components in isolation or combined. This advantage is at the expense of a control group, as none of the identified factorial trials reported to have a control condition. This does not necessarily have to be interpreted as a disadvantage, since the overall structure of factorial designs is different and each factor has its own control group consisting of a combination of conditions. As a consequence, a lower number of study participants need to be included [34].

#### 3.2.3. Sequential Multiple Assignment Randomised Trial

The study by Gonze et al. [32] was the only identified study which applied a SMART design. The study’s objective was to evaluate a smartphone app for physical activity. Like factorial designs, SMARTs are designed to evaluate multiple interventions and their responses. Gonze et al. [32] conducted a two stage SMART by first randomising participants to either group 1 (smartphone app only) or group 2 (smartphone app + tailored messages). Based on maintenance, increase, or decrease of steps, participants were categorised as either being responsive or non-responsive. The non-responsive participants were then re-randomised to the two existing groups (group 1 and 2) and a third new group who received a smartphone app, tailored messages, and gamification [32]. 

In contrast to the MRT design, where randomisation is done at the participant level, the SMART design follows a purposeful randomisation process. Based on the participants’ response behaviour (non-responder vs. responder), interventions are adjusted and in a further step randomly assigned. This also highlights the differences to RCTs. First, the SMART design is adaptive. While conducting the study, different treatment options are tested. By doing that, non-effective intervention components can be excluded. Second, a SMART achieves study results which are individualised to specific subgroups (responder vs. non-responder), thus answering the question of what works for whom. In contrast to factorial designs, SMARTs provide a control condition with which the adapted interventions are compared [32].

#### 3.2.4. Stepped-Wedge Cluster Randomised Trials

One stepped-wedge randomised trial could be identified in the search. The study by Palermo et al. [28] evaluated the effectiveness of a digital psychological intervention for paediatric chronic pain patients. For this study, participants were recruited from eight clinics in five different hospitals. The participants were randomised to one of four waves in which the groups crossed from a control condition to the intervention condition. The four waves represented different points in time when this crossover was performed. A control period was also present in which no clinic received the intervention. By using a random sequential crossover design, more and more participants received the intervention consisting of a mobile app to learn about pain self-management techniques [28].

Although this design is most similar to RCTs, it shows some differences. In this specific study of Palermo et al. [28], external validity was enhanced since intervention and control were conducted in real-world environments by using a sequential crossover design. Furthermore, intervention effectiveness and implementation data could be collected, which presented results that are more suitable for evaluating DHIs in their actual settings. In contrast, stepped-wedge designs differ from factorial designs and SMARTs, because they test the whole intervention and hence components cannot be compared with each other. 

## 4. Discussion

This is the first scoping review that has systematically addressed the question of which alternative evaluation methods are applied for DHIs. One main finding of this review is that RCTs are still used more often compared to other study designs that allow higher adaptivity in DHI studies. We identified four alternative study designs, but the studies were difficult to find in databases and most of them could not be identified by a systematic search syntax. 

Factorial designs were frequently used to evaluate DHIs. Stepped-wedge designs, SMARTs, and MRTs can still be considered as exotic study designs that have been insufficiently applied in research. MRTs and SMARTs can be helpful, as they allow for excluding ineffective intervention components in advance of testing the whole intervention. 

However, in addition to the described advantages, there are also limitations of the alternative study designs. In particular, the MRT and SMART approaches focus on the development and optimisation of intervention components. Several identified studies [25,29] point out that a RCT should be conducted afterwards to determine the effectiveness. Factorial designs are often performed within the MOST approach. The MOST approach aims to identify the most suitable combination of components within a DHI. Following the identification and refining process of these components, the approach also foresees a confirming phase in which an RCT is used for a comparison of the best adaptive intervention strategy against a suitable comparison group [36]. Accordingly, RCTs and alternative study designs often differ in their objectives. Therefore, it seems more adequate to label them as complementary rather than alternative designs. 

Alternative or complementary study designs also differ from RCTs in the use of health care data. They intend to use data directly from the health care setting rather than in strictly controlled study environments as it is the case with RCTs [37]. The increased focus on data collected in the real health care setting might be associated with a lack of controlled conditions in intervention and control groups. This do not necessarily need to be a limitation to these designs, because RCTs have been criticised for results lacking external validity [38]. This criticism can be addressed by upstream alternative study designs in which the truly effective intervention components are identified and tested in the real world and then confirmed under laboratory conditions in an RCT. However, this would conflict with the aim of making alternative study designs shorter and faster to conduct, as combining the two approaches would lead to even longer evaluation cycles. 

Another difference between MRTs as well as SMARTs and RCTs is the varying approach of random assignment. While RCTs assign participants to intervention or control group once, randomisation happens more frequently in MRTs and SMARTs. This approach may be criticized as detrimental to external and internal validity [39]. Thus, temporal variation of the same individual could introduce a bias effect on the testing of different intervention effects. Observed intervention effects might also not be independent of intervention components assigned at earlier decision points [37]. However, in our understanding, this is not the case, especially with SMART designs, as responders and non-responders are particularly subdivided to modify the intervention on this basis. This could also address the problem that physicians or scientists, for example, lack evidence-based decision rules for tailoring interventions or changing intervention components when patients or subject groups do not respond positively to the treatment. Alternative study designs might help to provide those groups with empirical guidance for developing personalized and effective interventions [40].

Gensorowsky et al. [37] describe in regard to SMART the deliberately increased tolerance of the type I error due to multiple measurements and its problems regarding specificity and the increased uncertainty of intervention effects. Nevertheless, the study design in the study by Gonze et al. [32] did not lead to an increase in the type I error, so this limitation of alternative study designs cannot be verified. Thus, there seems to be a consensus that alternative study designs automatically accept the increased tolerance of type I error. 

Even though most of the studies identified in this search implemented factorial designs to evaluate intervention effects, factorial experiments are still underrepresented in studies of intervention efficacy and effectiveness. This might be due to the fact that factorial designs examining several components as well as interactive effects at different time points can result in analytical challenges. When it comes to comparing one condition with another condition (i.e., control), RCTs are usually the best choice [41]. However, in a factorial design, the effectiveness of all individual components can be evaluated with the same power without explicit control, which leads to a lower overall sample size than in a traditional RCTs with multiple arms. Thus, based on a relatively small sample, the main effects and interactions of the intervention components can be estimated, which refer to all conditions and thus the overall sample [42].

Stepped-wedge designs are currently seldom used for measuring effectiveness. However, these study designs offer two key advantages that also need to be considered regarding DHIs. First, stepped-wedge designs can be used when it can be assumed that the intervention will not cause harm or it is unethical to withhold the intervention from participants. Second, they are also well suited when, for financial, logistical, or pragmatic reasons, the intervention can only be applied in stages [43]. Nevertheless, stepped-wedge designs differ from the identified study designs, because they already aim to measure the effectiveness of a fully developed intervention. This is an important difference to the factorial designs, SMARTs and MRTs, as these aim at identifying the most effective interaction of components. 

### 4.1. Comparison to Other Work

Comparing the results of this review with other conducted studies in this field of study, the results are comparable. Although the discussion about alternative study designs is getting more attention and it is stated that RCTs face certain challenges [10,14], especially when it comes to fast-moving and adaptable digital interventions, the majority of studies [44] seem to only recommend alternative evaluation designs without implementing them in practice. In the scoping review of Bonten et al. [11], the authors also concluded that although many alternative evaluation methods exist, these have not yet found their way into digital health research. DHIs are still predominantly evaluated using traditional evaluation methods. The screening process of this scoping review showed that many qualitative interviews [45,46] and feasibility studies [47] were used in the evaluation of DHIs. Often, participants in the studies were asked qualitatively or by using questionnaires about the usability or feasibility of DHIs. While this information also needs to be considered for the evaluation of DHIs as acceptance is also an important pillar of implementation [48], the evaluation of the effects cannot be done exclusively by such methods. Bonten et al. [11] made similar conclusions in their study asking eHealth researchers about their evaluation methods. The surveyed eHealth researchers indicated that their most common eHealth evaluation approach was feasibility or user questionnaire studies. Although the respondents agreed that the most appropriate approaches for evaluating eHealth applications were pragmatic RCTs or stepped-wedge trial designs, these approaches were not typically applied in evaluations [11]. 

Similar results were reported by Pham et al. [20], who also conducted a scoping review of alternative evaluation methods. They found that 80% of the identified studies were RCTs. They concluded that RCTs are still the gold standard for any clinical trial examining the efficacy of apps. A similar conclusion was drawn by Gensorowsky et al. [37] as they stated that neither MOST, SMART, nor MRT are primarily used for measuring efficacy, but for intervention development and optimisation only. The authors criticised that these study designs deviate strongly from methodological procedures (statistical error tolerance or randomisation procedure) of RCTs and therefore hold serious biases. Hence, the authors disapprove these alternative study designs as a surrogate for RCTs [37].

Currently, those alternative concepts are scarce due to multiple reasons. Structural requirements are strongly oriented towards traditional evaluation methods. Thus, studies with alternative concepts, such as MOST and SMART, have difficulties receiving funding for conducting the evaluation, as funding agencies and reviewers are often unfamiliar with these concepts. In addition, there is also the question of how alternative study designs can be pre-registered, although the registries are predominantly designed for RCTs. 

As mentioned above, the screening of studies identified a high number of qualitative studies. Although these were excluded in this scoping review, they should be included in any holistic and continuous evaluation. An increasingly considered mixed methods approach for a comprehensive evaluation is the realist evaluation. The realist evaluation methodology was developed by Pawson and Tilley [49,50] and is a procedure for testing a theoretical mechanism of change in complex interventions. Interventions are considered in their real-life implementation and not only tested for effectiveness under laboratory or otherwise standardised conditions. While RCTs only ask about the effectiveness of an intervention, the realist evaluation provides information about impact pathways in complex systems. Realist evaluations address the question “What works for whom, under what conditions, and how?” Thus, the approach would also be suitable to decipher the impact of components and its interaction with contextual influences of DHIs [51]. 

In addition to the question of how to evaluate DHIs best, evaluation should be considered during development, pilot testing, evaluation, and post testing as an evaluation circle. Bonten et al. [11] and Enam et al. [52] suggested the generation of robust evidence at all stages. As interventions become more complex, it is not sufficient to rely on one method of evidence (e.g., RCT). Rather, different methods must be used in different phases. It remains an open question whether the evaluation of DHIs should be analysed in phases or whether continuous evaluation is more appropriate. Future research also needs to address the question of whether we need more complex study designs for DHIs or whether we need to rethink the approach to evaluation of DHIs. Collins et al. [34] proposed, e.g., factorial designs as methods for identifying the best possible component mix of a DHI, which are then tested through traditional evaluation designs such as RCTs. Accordingly, the evaluation process of DHIs becomes more complex, which, however, might also allow to determine the effects of DHIs more precisely. Many questions are still unanswered in this field and therefore this review aimed to provide a basis for the discussion of further evaluation methods to generate robust evidence of DHIs in addition to RCTs [11,52].

### 4.2. Strengths and Limitations

This scoping review has some limitations that must be accounted for when interpreting the results. A limited selection of databases was searched. We searched major databases, which can be regarded as a strength, but although numerous studies were screened, relevant studies might have been missed. Alternative study designs studies were especially difficult to identify, which explains the high number of studies that were included by hand search. However, only peer-reviewed articles were included, which increases the quality of the scoping review. Another limitation is the impossibility of conducting a systematic review or meta-analysis due to the relative new field of evaluation practice and the different approaches. The methodology of a scoping review was perceived as more suitable for the aim of providing an overview over a large and diverse body of literature [23]. In such a design, different alternative study designs were descriptively presented, but the material was not appraised for quality. Furthermore, since our review was developed to provide only an overview of alternative evaluation methods currently in use, it does not allow for an assessment of the appropriateness of alternative study designs to prove the effects of DHIs.

## 5. Conclusions

This review identified four alternative study designs in eight different trials and discussed potential advantages concerning the use of these study designs. It can be concluded that DHIs are mainly still evaluated by traditional evaluation methods, despite some shortcomings regarding their suitability for the evaluation of DHIs. This scoping review constitutes a basis for discussion as it is critically assessing the current evaluation methods for DHIs and depicts some research gaps, demonstrating a need for further research concerning how to best evaluate DHIs. A clear recommendation regarding the best method for evaluating DHIs cannot be given based on the findings of this review. However, it should be noted that DHIs are often developed as complex interventions that will benefit from evaluation approaches that are tailored to the specific needs of the intervention, which often exceed the scope of RCTs. It needs to be evaluated in further studies whether alternative designs can achieve a similar robustness of results in comparison to RCTs.

## Figures and Tables

**Figure 1 ijerph-19-05221-f001:**
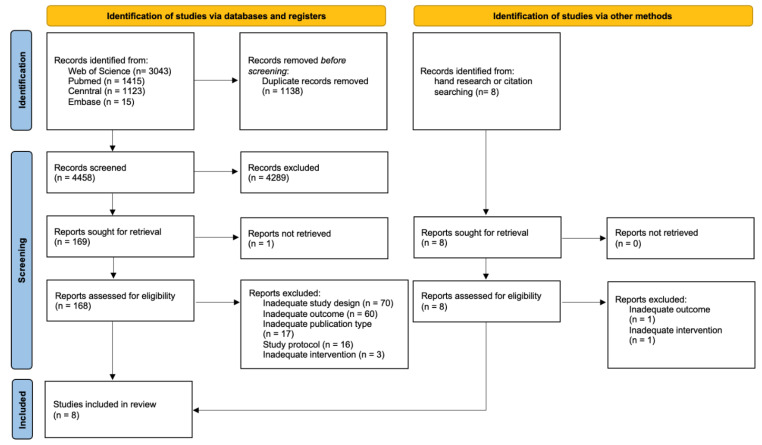
PRISMA 2020 flow diagram of literature search and selection process [24].

**Table 1 ijerph-19-05221-t001:** Search Strategy.

Search Terms	Synonyms
Digital Health	Electronic Health, EHealth, Mobile Health, MHealth, Digital Health, Telehealth, Health Technology
Evaluation methods	Summative evaluation, Evaluation Studies as Topic (MeSH), Evaluation Methods, Alternative Study Designs, Evaluation Study (MeSH), effective*, efficacy, trial, “Research Design” “Randomised Controlled Trials as Topic/methods”, “Evaluation Studies as Topic”, Research Method*[tiab], Research Strateg*[tiab], Methodolog*[tiab], Alternative*[tiab] Effective*[tiab], Evaluation*[tiab], Quality[tiab])

**Table 2 ijerph-19-05221-t002:** Study Characteristics of the Included Studies.

Author (Year) & Country	Study Design	Study Purpose	Study Sponsor	Targeted Condition	Data Collection Time-Points (Amount)	Duration (Weeks)	Sample Size	Control Group (CG) & Intervention Group (IG)	Masking	Group Assignment
Klasnja et al. (2019) [25] United States	MRT	Evaluation of efficacy of activity suggestions	Public funding	Physical activity	Daily (7540)	6	44	CG: None IG: Tailored walking suggestions	Participant: N.A. Practitioner: N.A. Assessors: N.A.	At each decision point: Individual randomisation to either no suggestion, walking suggestion or anti-sedentary suggestion
Adams et al. (2017) [26] United States	Factorial 2 × 2 design	Evaluation of effects for goal setting and rewards to increase daily steps	Public funding	Physical activity	Baseline and 4-months follow-up (2)	16	96	CG: None IG: Four intervention components (adaptive vs. static goal setting and immediate vs. delayed rewards)	Participant: None Practitioner: N.A. Assessors: Yes	Individual randomisation to one of four intervention components after baseline
Gonze et al. (2020) [32] Brazil	SMART	Evaluation of effects of a smartphone app for physical activity	Public funding	Physical activity	Baseline, 12-week follow-up and 24-week follow- up (3)	24	18	CG: TAU IG: Three intervention components (app only, app + tailored messages, and app + tailored messages and gamification)	Participant: None Practitioner: N.A. Assessors: Yes	First stage intervention: Individual randomisation to Group 1 (app only), Group 2 (app + tailored messages) or control group Second stage intervention: Individual rerandomisation of non-responders to Group 1 or 2 or Group 3 (app + tailored messages and gamification)
Du et al. (2016) [27] United States	Factorial 2 × 2 design	Evaluation of effects of a mHealth application on eating behaviour, physical activity, and stress level	Public and private funding	eating behaviour, physical activity, and stress level	Baseline, pre-test, and post-test follow-up (3)	8	124	CG: TAU IG: Four intervention conditions (emailed wellness programme, emailed wellness programme + team support, mobile walking and stress intervention, and mobile walking and stress intervention + team support)	Participant: Yes Practitioner: N.A. Assessors: N.A.	Individual randomisation to one of four intervention components before baseline
Palermo et al. (2020) [28] United States	Stepped-wedge cluster randomised trial	Evaluation of effectiveness and implementation of a digitally delivered psychosocial intervention for paediatric chronic pain	Public and private funding	Paediatric chronic pain	Baseline, 8ƒ-week follow-up and 3-month follow-up (3)	20	143	CG: TAU IG: Self-guided smartphone app for patients and their parents	Participant: None Practitioner: None Assessors: Yes	Random sequential crossover of the clinics in 1 of 4 waves from control to intervention
Schroé et al. (2020) [31] Belgium	Factorial 2 × 2 × 2 design	Evaluation of efficacy of behaviour change techniques on physical activity and sedentary behaviour	Public funding	Physical activity and sedentary behaviour	Baseline and 5-week follow-up (2)	5	473	CG: No behavioural technique IG: Seven intervention conditions consisting of action planning, coping planning, and self-monitoring	Participant: Yes Practitioner: None Assessors: N.A.	Block randomisation of participants to one of eight (control group counted in here) intervention groups
Spring et al. (2020) [29] United States	Factorial 2 × 5 design	Identification of intervention components that enhanced weight loss	In part Public Funding	Weight	Baseline, 3-months follow-up and 6-months follow-up (3)	24	562	CG: None IG: 32 intervention conditions consisting of coaching calls, primary care provider reports, meal replacements, buddy training, and text messaging	Participant: None Practitioner: None Assessors: Yes	Block randomisation of participants to one of 32 intervention groups
Strecher et al. (2008) [30] United States	Fractional factorial 2 × 4 design	Identify intervention components of a web-based smoking cessation programme	Public funding	Smoking	Baseline and 6-months follow-up (2)	24	1866	CG: None IG: 16 intervention conditions consisting of tailored success story, outcome expectation, efficacy expectation messages, source personalization, and exposure	Participant: Yes Practitioner: N.A. Assessors: N.A.	Individual randomisation to one of 16 intervention components

MRT: Micro Randomised Trial, CG: Control Group, IG: Intervention Group; N.A.: Not Available, TAU: Treatment As Usual, SMART: Sequential Multiple Assignment Randomised Trial.

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
