# Peer review of "Evaluation Methods Applied to Digital Health Interventions: What Is Being Used beyond Randomised Controlled Trials?—A Scoping Review"

_ijerph, 2022, doi:10.3390/ijerph19095221_

Round 1
Reviewer 1 Report
This study reviewed evaluation methods in the digital health intervention (DHIS) literature. It aims to find alternatives to randomized controlled trials and to classify more appropriate evaluation methods. In this study, eight different trials were extracted, four alternative study designs were identified and compared with the gold standard randomized controlled trial RCTS to discuss the advantages and disadvantages of each study method. But there are some limitations that need further improvement.
Introduction
What is the theoretical basis for the "stepped-wedge design or N-of-1 design has been extended to include research designs grouped under the heading of alternative research designs"?
Methods
- Whether the 176 selected eligible studies have been evaluated for quality to ensure the authenticity and reliability of conclusions? Such as, the internal authenticity of bias error processing, and whether it can be popularized with the external authenticity?
- Please describe the screening criteria of studies in detail.
Results
Why were the eight studies included in the scoping review? Please describe the reasons.
Discussion
- Please explain specifically how factorial designs were used to evaluate DHIs?
- Theinterventions were screened according to the characteristics of the specific study designs and their difference to RCTs in this study, but the accuracy of alternative assessment therapies cannot be guaranteed due to lack of standards to evaluate the applicability of examples of each alternative therapy.
Conclusions
Please provide some advice on how to best evaluate DHI.
Author Response
Point 1: What is the theoretical basis for the "stepped-wedge design or N-of-1 design has been extended to include research designs grouped under the heading of alternative research designs"?
Response 1: Thank you for this question. We changed the paragraph and hope that this explanation now makes clear that both study designs were proposed by the MRC as alternatives for the evaluation of interventions. We have implemented the following statement:
The British Medical Research Council proposed extending the range of study designs for evaluation beyond RCTs and mentioned e.g. stepped-wedge designs or N-of-1 designs that can be considered as alternative options [14]. These alternative options for the evaluation of interventions were summarized under the heading of "alternative study designs". The definition alternative study designs was also adopted from other authors [15] (p. 2, line 70-74).
Point 2: Whether the 176 selected eligible studies have been evaluated for quality to ensure the authenticity and reliability of conclusions? Such as, the internal authenticity of bias error processing, and whether it can be popularized with the external authenticity?
Response 2: Thank you for this comment. For this manuscript, the method of a scoping review was chosen. In preparing the scoping review, explicit reference was made to Arksey & O'Malley (2005). According to the two authors, a scoping review does not assess the quality of the studies. This distinguishes it from a systematic review. We aimed to provide an overview of the methodologies used and did not assess the conclusions made in the selected publications.
We have pointed this out in the limitations section (“In such a design, different alternative study designs were descriptively presented, but the material was not appraised for quality.” (p. 16, line 444-445)).
In addition, we have added a sentence in the methods section:
“Arksey and O’Malley [21] methodology framework for scoping reviews guided the conduct of this review and following this framework no quality assessment of the studies was intended.” (p. 3, line 94-97).
Point 3: Please describe the screening criteria of studies in detail.
Response 3: Thank you for this remark. We have included the following in-depth information to describe the screening process in more detail:
“If the title and abstract of the respective study did not seem appropriate for the research question, it was excluded. If there was uncertainty or a final assessment of the study was not possible, the study was included to avoid excluding important studies in the first screening phase. Second, full texts of all records that were deemed relevant after the first screening phase were obtained and reviewed by two authors (RH, SMH) for eligibility.” (p. 4, line 142-145).
Point 4: Why were the eight studies included in the scoping review? Please describe the reasons.
Response 4: Thank you. We have added the following sentence to clarify this point:
“(…) eight studies were included in the scoping review that met all pre-defined inclusion criteria but were not disqualified based on exclusion criteria during the iterative screening phase.” (p. 4, line 160-161).
Point 5: Please explain specifically how factorial designs were used to evaluate DHIs?
Response 5: Thank you for pointing this out. The exact characteristics of factorial designs and how they were used are formulated in the results section under 3.2.2. (p. 12-13, line 224-253).
Point 6: The interventions were screened according to the characteristics of the specific study designs and their difference to RCTs in this study, but the accuracy of alternative assessment therapies cannot be guaranteed due to lack of standards to evaluate the applicability of examples of each alternative therapy.
Response 6: Thank you for this advice, this is correct. The aim of our study was to identify alternative study designs that were already used and reported as a consequence of many narrative overviews stating that alternative study designs are an important option especially when it comes to the evaluation of DHIs. However, we cannot make any conclusions about the suitability of these studies to identify effects. Therefore, we mentioned this now in the limitations by the statement:
“Furthermore, since our review was developed to provide only an overview of alternative evaluation methods currently in use, it does not allow for an assessment of the appropriateness of alternative study designs to prove effects of DHIs.” (p. 16, line 445-448).
We agree with you that comparing the accuracy of alternative designs to RCTs would be an interesting next step for future research.
Point 7: Please provide some advice on how to best evaluate DHI.
Response 7: We have added an advice in the conclusions section and also added a statement about the robustness of findings of alternative designs:
“A clear recommendation on the best method for evaluating DHIs cannot be given based on the findings of this review. However, it should be noted that DHIs are often developed as complex interventions that will benefit from evaluation approaches that are tailored to the specific needs of the intervention, which often exceed the scope of RCTs. It needs to be evaluated in further studies, whether alternative designs can achieve a similar robustness of results in comparison to RCTs.” (p. 16, line 461-466).
References:
Arksey, H., & O'Malley, L. (2005). Scoping studies: towards a methodological framework. International journal of social research methodology, 8(1), 19-32.
Reviewer 2 Report
This study analyzed the characteristics and usefulness of various evaluation methods for DHI through literature research. As a 'Review' type of work, many types of literature were analyzed and systematically classified.In order to improve the completeness of this study, please revise the following.
- Please describe DHI in more detail in Chapter 1. In other words, please describe the purpose, usage, and development history of DHI.
-In Chapter 2, describe in more detail how detailed criteria for inclusion criteria and exclusion criteria were developed.
- In 4.2. Limitations, the first disadvantage, i.e., research on only major databases and major journals, is seen as an advantage, not a disadvantage. Therefore, the first drawback is to be eliminated.
Author Response
Point 1: Please describe DHI in more detail in Chapter 1. In other words, please describe the purpose, usage, and development history of DHI.
Response 1: Thank you for this advice. We have added the following paragraph:
“DHIs are characterised by their ability to support and serve different health needs of providers, patients and populations formally or informally through digital technologies. The application and use of DHIs is diverse and ranges from simple SMS support to complex modular interventions that can be used as an app for doctors, patients or entire populations [4].“ (p. 1, line 37-41)
Point 2: In Chapter 2, describe in more detail how detailed criteria for inclusion criteria and exclusion criteria were developed.
Response 2: Thank you for this important notice. We have added a sentence to clarify the development process of the inclusion and exclusion criteria:
“The inclusion and exclusion criteria were determined a priori in a consensus process by the researchers conducting the review (SH and RH).” (p. 3, line 104-105)
Point 3: In 4.2. Limitations, the first disadvantage, i.e., research on only major databases and major journals, is seen as an advantage, not a disadvantage. Therefore, the first drawback is to be eliminated.
Response 3: Thank you very much for this comment. We have now formulated the section more balanced:
“Strengths and Limitations
This scoping review has some limitations that must be accounted for when interpreting the results. A limited selection of databases was searched. We searched major databases, which can be regarded as a strength, but although numerous studies were screened, relevant studies might have been missed.” (p. 16, line 439-442)
Reviewer 3 Report
Great topic!
All the paper is very well written and the aim is clear. There are no clear mistakes, the search strategy is adequate, all the results and discussion are well presented. The authors should be appraised for such a good work.
I will just point some questions, for further reflection, that the authors might want to include in their discussion: As the authors note, these new evaluation methodologies are more directed at the development stage of an intervention (line 286-292).
So, are we looking to more complex study designs of the evaluation, when what we really need is to improve the design of the intervention before we set out to evaluate it with a “conservative” methodology such as an RCT? I think the paper would improve if some discussion around this topic could be included.
Author Response
Point 2: I will just point some questions, for further reflection, that the authors might want to include in their discussion: As the authors note, these new evaluation methodologies are more directed at the development stage of an intervention (line 286-292). So, are we looking to more complex study designs of the evaluation, when what we really need is to improve the design of the intervention before we set out to evaluate it with a “conservative” methodology such as an RCT? I think the paper would improve if some discussion around this topic could be included.
Response 2: Thank you very much for your supporting and very good feedback. We have added the following to Chapter 4.1.: “Future research also needs to address the question of whether more complex study designs for DHIs are needed or whether the approach to evaluation of DHIs needs to be rethought. Collins et al. [32] proposed e.g. factorial designs as methods for identifying the best possible component mix of a DHI, which are then tested through traditional evaluation designs such as RCTs. Accordingly, the evaluation process of DHIs becomes more complex, which, however, also might allow determining the effects of DHIs more precisely.” (p. 16, line 427-433)